# The Credibility of Health Information Sources as Predictors of Attitudes toward Vaccination—The Results from a Longitudinal Study in Poland

**DOI:** 10.3390/vaccines9080933

**Published:** 2021-08-23

**Authors:** Katarzyna Stasiuk, Mateusz Polak, Dariusz Dolinski, Jozef Maciuszek

**Affiliations:** 1Institute of Applied Psychology, Jagiellonian University, 31-007 Krakow, Poland; mateusz.polak@uj.edu.pl (M.P.); jozef.maciuszek@uj.edu.pl (J.M.); 2Department of Psychology in Wroclaw, SWPS University of Social Sciences and Humanities, 03-815 Warsaw, Poland; ddolinsk@swps.edu.pl

**Keywords:** attitude towards vaccination, trust in physicians, trust in science, Internet as the source of health information, COVID-19

## Abstract

Background: The research focused on the relationships between attitudes towards vaccination and the trust placed in different sources of information (science, experts and the information available on the Internet) before and during COVID-19. Method: A longitudinal design was applied with the first measurement in February 2018 (*N* = 1039). The second measurement (*N* = 400) was carried out in December 2020 to test if the pandemic influenced the trust in different sources of information. Results: The final analyses carried out on final sample of 400 participants showed that there has been no change in trust in the Internet as a source of knowledge about health during the pandemic. However, the trust in science, physicians, subjective health knowledge, as well as the attitude towards the vaccination has declined. Regression analysis also showed that changes in the level of trust in physicians and science were associated with analogous (in the same direction) changes in attitudes toward vaccination. The study was also focused on the trust in different sources of health knowledge as possible predictors of willingness to be vaccinated against SARS-nCoV-2. However, it appeared that the selected predictors explained a small part of the variance. This suggests that attitudes toward the new COVID vaccines may have different sources than attitudes toward vaccines that have been known to the public for a long time.

## 1. Introduction

Despite overwhelming medical evidence and the unanimous position of medical professionals in favor of vaccination, the number of people skeptical about vaccination has grown in many countries in recent years. The consequence of this alarming trend is a decline in immunization coverage [1]. The problem of expanding vaccine rejection has begun to be so widespread that it became the subject of many studies aimed at understanding the sources and correlations of attitudes toward vaccination. Their results have shown that vaccine hesitancy may be related to individual factors such as religious orthodoxy, individualistic/hierarchical worldviews or conspiratorial thinking [2,3].

In our study, we approached the problem of attitudes toward vaccination not from the perspective of personality traits, but by treating it as a manifestation of the broader problem of trust in science and experts’ opinions [4,5,6]. This issue has become particularly relevant at this time, when the world is struggling to cope with the problems caused by the COVID-19 pandemic, which, according to many experts, can only be contained through mass vaccination. We also took into account that faced with the pandemic, laypeople may have changed their levels of trust in various sources of health knowledge [7]. This in turn might have influenced their attitude toward vaccination, including toward the vaccines against SARS-nCoV-2. In February 2018, we conducted a survey on attitudes toward vaccination and different sources of health knowledge. We decided to reach out to the participants again in December 2020 to see how the pandemic had changed their beliefs, and whether these beliefs were also related to their decision on being vaccinated against COVID-19.

### 1.1. Trust in Physicians and Science as Predictors of Vaccine Attitudes before and during the Pandemic

A lot of data show that advice from healthcare experts is the most influential source of information about vaccination for most people [8]. However, in recent years the trust in physicians has declined, which has also affected attitudes toward vaccination. In the “pre-internet” era, physicians used to be the main source of reliable health information, which strengthened the unquestionability of their opinions [9,10]. Widespread access to the Internet has made information about health easy to find and assimilate, and laypeople have also gained additional sources of information about other patients. In comparison with their experiences and beliefs (including regarding vaccinations), even the advice of the most respected vaccine authority sometimes “becomes just another opinion” [11]. All of these factors have undermined the hitherto unquestioned authority of physicians, including in influencing attitudes toward vaccination.

The downturn in trust in vaccines may be a manifestation not only of declining trust in physicians, but also of a broader social trend of diminishing trust in science [12]. While some people respect science and scientists, others are skeptical of them [13,14,15]. When it comes to scientific work in the medical field, some individuals think that scientists are ‘under the thumb’ of pharmaceutical companies and suspect them of putting profits above public interest, [5,16]. These doubts, increasingly present in the public sphere in recent years, have strongly undermined confidence in vaccination. However, this situation may have changed as a result of the COVID-19 pandemic. For the past year, the rapidly increasing number of cases and deaths caused by SARS-nCoV-2 and the race against time to develop a vaccine for the new virus have caused many eyes to turn to healthcare providers, researchers and science, and affected attitudes toward them. The evidence from the previous studies is ambiguous, showing that trust in physicians and science increased in some countries during the pandemic but decreased in others [17,18]. Attitudes toward vaccination (also toward the new vaccines for COVID-19) are similarly diverse. Although the number of people wanting to be vaccinated in many countries is high and the situation looks increasingly positive on a global scale, the results for some individual countries paint a more complicated picture—in particular those that have a history of vaccine hesitancy (e.g., France or Poland) [19].

Considering the above, our goal was to investigate whether the attitudes toward vaccination (in general and against COVID-19 in particular) may stem from the trust in physicians and/or anti-scientific attitudes. Moreover, we wanted to investigate how the COVID-19 pandemic has influenced these attitudes.

### 1.2. Trust in the Internet and Subjective Health Knowledge as Predictors of Attitudes toward Vaccination before and during the Pandemic

Over the past few decades, the Internet has become an important source of medical information for patients. People often visit websites to explain or diagnose their physical ailments; they also consider the Internet an easily accessible source of information about vaccinations [20,21]. However, research on the content of health websites has highlighted inaccuracies that raise concerns about the quality of online health information. The limited accuracy of information is often a result of the Internet’s most distinctive feature, which is that anyone can potentially publish health information [22]. Moreover, in recent years, the Internet has become an ‘ally’ of anti-vaccine movements, allowing their arguments to be promoted on an unprecedented scale. A recent report by the Centre for Countering Digital Hate has found that social media accounts run by vaccine skeptics have increased in followers by at least 7–8 million people since 2019 [23]. The problem has become even more serious over the past year, as the anti-vaccine movement saw the pandemic as a great opportunity to create content; therefore, people searching for COVID-19-related information would find anti-vaccine arguments. In addition, vaccine deniers see this also as an opportunity to increase parents’ doubts about routinely vaccinating their children [24].

Taking the above considerations into account, we included trust in health-related information presented on the Internet as a possible predictor of attitudes toward vaccination (including vaccination against COVID-19). We also wanted to investigate whether the pandemic would change this level of trust and its impact on the attitudes toward vaccination (compared to in 2018).

In our study, we also hypothesized that attitudes toward vaccination may be related to beliefs about one’s own health-related knowledge. We based these assumptions on previous research into subjective knowledge. Research on judgments and decision making has a long tradition of distinguishing between objective knowledge and subjective knowledge [25,26]. Objective knowledge is information stored by an individual in their long-term memory [25]. Subjective knowledge, in turn, refers to the self-perception of one’s own knowledge, and includes a level of confidence in an individual’s own perceived knowledge [27,28]. While it is reasonable to assume that people’s perception of their subjective knowledge strongly reflects their actual knowledge in a given domain, a lot of research has shown that people tend to overestimate it even in the contexts in which they are not educated or professionally engaged [29]. This overconfidence may have important consequences—people who have high subjective knowledge in a given domain express less trust in experts and are less likely to follow their recommendations. This phenomenon is clearly visible in the health domain. One of the previous studies related to vaccination attitudes showed that people who believed that their knowledge about autism was as high as experts’ knowledge exhibited more reservations toward mandatory childhood vaccination [30].

Although many studies show that people tend to positively evaluate their own knowledge in various areas, the doubts and uncertainty associated with a pandemic may have changed these optimistic beliefs. Therefore, in the second measurement we decided to repeat the questions about the assessment of the participants’ own knowledge in the area of health. We also wanted to determine whether assessing one’s own knowledge would prove to be a significant predictor of attitudes toward vaccination.

To sum up all the above considerations, our research goal was to test the trust in physicians, science, the Internet and one’s own knowledge as predictors of attitudes toward vaccination, before and during the COVID-19 pandemic. We also applied a longitudinal design to investigate whether the attitudes toward the sources of health information and toward vaccines changed after the breakout of the pandemic or remained relatively stable.

## 2. Materials and Methods

### 2.1. Participants (Demographic Characteristics)

A representative sample of the Polish general population (*N* = 1039) took part in the first measurement (February 2018). The study was run online by the Ariadna Nationwide Research Panel, a Polish counterpart of mTurk—a company specialized in the polling of large samples for the purpose of research. The panel enables random selection of a sample from among 300,000 registered and verified persons. The socio-demographic profile of the persons registered on the panel corresponds with the profile of Polish internet users. Additionally, Ariadna has been awarded certificates issued by recognized organizations associated with social research companies (including ESOMAR). For participation in the survey, respondents received credit points that they could exchange for gifts.

The representative sample was drawn from over 300,000 active Ariadna participants. A random quota sampling method was used, based on sex (2 subgroups), age (5 subgroups and place of residence (5 subgroups), each demographic criterion controlled to be representative of the Polish general population, giving a total of 50 weighted cells. Weights were calculated based on these three demographic criteria, and participants were drawn randomly from cells to fit demographic quotas. Because we decided to apply a longitudinal design, we tried to reach the same group of respondents in December 2020. Reaching the same sample for a second time was possible as Ariadna uses unique identifiers for their respondents and there is a possibility to reach out to particular respondents inviting them for a given survey.

However, in the second measurement we were able to reach only some of the participants from this group, and we finally conducted analyses on the individuals who participated in both the first and second measurements. Therefore, the final sample was *N* = 400, 175 women and 225 men. A total of 24% were aged 18–34, 31% were aged between 35–54 and 45% were aged 55 or older. Moreover, 12% had primary or vocational education, 45% had secondary or postsecondary education and 43% had a bachelor’s degree. The proportion of men to women was significantly higher in this final sample than in the original one, and there was an age difference signifying a systematic dropout of the younger population. The sample retained its structure regarding residence and education.

### 2.2. Instrument

Questionnaires were presented online on the same survey platform in 2018 and 2020. The questionnaire consisted of 21 single-choice items which required on average 10 min. to be completed, and it was divided into six main categories: (a) demographic data, including age, gender, education, place of residence; (b) trust in physicians was measured with three statements (I trust the doctors’ knowledge; Treatment recommended by doctors is usually effective; Physicians’ recommendations are usually based on reliable knowledge). The responses to statements were measured on an 11-point Likert scale from 0 (strongly disagree) to 10 (strongly agree); (c) subjective health knowledge was measured with four questions (I am very knowledgeable in health-related matters; Sometimes I know better than physicians what will be good for my health; I often feel like I know more about health than others; I usually know what is best for my health). The responses to the statements were measured on an 11-point Likert scale; (d) trust in science was measured with four statements (Science allows us to understand the world better than religion; Science shows us the true picture of the world and people; Only through scientific knowledge can the most important problems of people and the world be solved; Scientific knowledge must be trusted). These statements were based on Farias and Reiman’s [31] Belief in Science scale. The responses to the statements were measured on an 11-point Likert scale; (e) trust in the Internet as a source of health knowledge was measured with four questions (Before I go to the doctor I look online for information about my symptoms and try to figure out what’s wrong with me; Before I start taking drugs prescribed by my doctor I prefer to check online what other patients think about them; I often have more confidence in health information found online than in the opinions of the doctors I visit; I trust the health advice that is available online). The responses to statements were measured on an 11-point Likert scale; (f) attitude toward vaccination was measured with two questions (Vaccines are an effective way to prevent the spread of infectious diseases; Children’s vaccines often do more harm than good for children). The responses to statements were measured on an 11-point Likert scale. In the second measure (December 2020) we also included a question about willingness to be vaccinated against SARS-nCoV-2 (Will you get vaccinated for coronavirus?) on a 5-point Likert scale (0—definitely not, 5—definitely yes).

### 2.3. Ethical Considerations

The study protocol was reviewed and approved by the Ethics Committee of the Institute of Applied Psychology, Jagiellonian University in Krakow (Poland). The questionnaire collected no identifying personal data from the participants.

### 2.4. Data Analysis

The IBM SPSS Statistics (SPSS) software version 26 was used to perform the statistical tests—repeated measures ANOVA and multiple linear regression.

## 3. Results

### 3.1. Changes in Trust toward Different Sources of Health Information and in Attitudes toward Vaccination between February 2018 and December 2020—Results of a Repeated-Measures ANOVA

In the first step of the analyses, we applied a Repeated-Measures ANOVA to assess the possible difference in trust toward different sources of health information and in attitudes toward vaccination between February 2018 and December 2020. The results revealed that mean scores for trust in physicians, trust in science, subjective knowledge and attitudes toward vaccination significantly decreased across the two time points, showing a significant time effect for these factors (Table 1). Trust in the Internet as the source of health information did not significantly change across the two measurements.

### 3.2. Trust in Different Sources of Health Information and Its Changes as Predictors of Attitudes towards Vaccination in February 2018 and December 2020—The Results of Regression Analyses

In the next step of the analyses, we conducted a separate multiple linear regression analysis for two measurements (in February 2018 and in December 2020), with attitudes towards vaccination as the dependent variable, and four independent variables: trust in physicians, trust in science, trust in the Internet as source of health information and subjective health knowledge (Table 2).

For the first measurement (February 2018), the regression model for attitudes towards vaccination explained about 34% of the variance. The trust in physicians, trust in science and trust in the Internet were the significant predictors of attitudes towards vaccination. The trust in physicians and trust in science had positive regression weights, showing that the higher the trust towards both these sources, the more positive is the attitude towards vaccination. The trust in the Internet had negative weights, which demonstrates that the more people that believe in the Internet as a reliable source of information about health, the less positive is their attitude towards vaccination. In the first measurement subjective knowledge did not contribute significantly to attitudes towards vaccination.

For the second measurement (December 2020), the regression model for attitudes towards vaccination explained about 49% of the variance. In this measurement, the four factors (including subjective knowledge) significantly predicted the attitudes towards vaccination. The trust in physicians and trust in science again had positive regression weights showing that the higher the trust towards both these sources, the more positive is the attitude towards vaccination. Trust in the Internet and subjective knowledge had negative weights. It demonstrates that the more people believe in the Internet as a reliable source of information about health and the higher they evaluated their health knowledge, the less positive is their attitude towards vaccination.

In this research, we also wanted to investigate if the changes in trust towards different sources of health information across two time points (February 2018 and December 2020) influenced the changes in attitudes towards vaccination across these two time points. In this purpose, we created rates of change for the dependent variable and for the three predictors by subtracting the result of measurement one from the result of measurement two. We did not create an indicator of change for trust in the Internet because the Repeated-Measures ANOVA showed that it did not change significantly over the two measurements. We then conducted a regression analysis, with changes in attitudes towards vaccination as the dependent variables, and three independent variables: changes in trust in physicians and in science, as well as changes in subjective health knowledge (Table 3).

The regression model for changes in attitudes towards vaccination explained about 17% of the variance. The changes in trust in physicians and trust in science, as well as the changes in subjective knowledge were the significant predictors of changes in attitudes towards vaccination. The changes in trust in physicians and trust in science had positive regression weights. That is, changes in trust toward these sources in a particular direction were associated with changes in attitudes toward vaccination in the same direction (e.g., a decline in confidence toward physicians predicts decreases in positive attitude toward vaccination). On the contrary, the changes in subjective knowledge had negative weights. It demonstrates that changes in these factors in a particular direction were associated with changes in attitudes toward vaccination in the opposite direction (e.g., a decrease in subjective knowledge predicts an increase in positive attitudes toward vaccination)

### 3.3. Trust in Different Sources of Health Information and Its Changes as Predictors of Attitudes Vaccination against SARS-nCoV-2 in December 2020—The Results of Regression Analyses

In the third step of our analyses, we conducted a separate multiple linear regression analysis for the second measurement (in December 2020), with attitudes towards vaccination against SARS-nCoV-2 as the dependent variable, and four independent variables: trust in physicians, trust in science, trust in the Internet as source of health information and subjective health knowledge (Table 4).

The regression model for attitudes towards vaccination against SARS-nCoV-2 explained about 12% of the variance. The trust in physicians and trust in the Internet were the significant predictors of attitudes towards vaccination against SARS-nCoV-2. The factors had positive regression weights showing that the higher the trust towards both these sources, the more positive is the attitude towards vaccination. The trust in science and subjective knowledge did not contribute significantly to attitudes towards vaccination against SARS-nCoV-2.

In the last step of the analysis, we also wanted to investigate if the changes in trust towards different sources of health information across two time points (February 2018 and December 2020) influenced the attitude towards vaccination against SARS-nCoV-2. We then conducted a regression analysis, with attitude towards vaccination against SARS-nCoV-2 as the dependent variable, and four independent variables: changes in trust in physicians, in science and in the Internet, as well as changes in subjective health knowledge. However, the percentage of explained variance appeared to be very low (0.2%); therefore, we did not present specific results.

## 4. Discussion

In this research, we focused on the influence of trust in physicians, science and the Internet as well as subjective knowledge on attitudes toward vaccination. Our second aim was to investigate the possible changes in the aforementioned factors across two points in time—February 2018 and December 2020, during the second wave of the pandemic. In addition, we wanted to test whether trust in various possible sources of information about health and its changes over time was related to willingness to be vaccinated against SARS-nCoV-2.

The results of the study showed that trust in physicians, science and the Internet were strong predictors of the attitudes toward vaccination at two points in time. The higher the trust in doctors, the more positive the attitudes toward vaccination, and conversely, lower trust is associated with higher skepticism toward vaccination. These results confirm the evidence obtained in many previous studies, showing that trust in physicians has a positive effect on vaccination intention and the provision of information by health professionals and the quality of their information are essential for the decision of whether to be vaccinated [32,33]. However, the problem is that our results also showed that trust in physicians has declined from 2018 to 2020, which appears to be related to a decline in attitudes toward vaccination.

Perhaps the decrease in trust in physicians is a result of healthcare problems faced by many countries (including Poland) during the pandemic. The results of public opinion polls show that many people are not satisfied with healthcare during the pandemic. It is difficult to arrange a traditional surgery visit, regardless of the specialization of the physician, and many patients do not trust e-visits or telehealth consultations, which cannot replace direct contact with a healthcare provider. Appointments with specialists are often subject to long waiting times and access to tests, even basic ones, is difficult [34].

Our results showed that between the years in which we conducted the studies, trust in science also declined, which was again associated with a decline in positive attitudes toward vaccination. In our view, this result can be explained in several non-exclusive ways. Perhaps this is due to an observation that scientists appeared for a long time to be losing the battle against the pandemic due to insufficient knowledge, and the belief that they failed to quickly identify and control the spread of the coronavirus. Perhaps the sense of one’s own helplessness, ignorance and uncertainty in the face of the pandemic also caused a decline in subjective health knowledge (which appeared to be the weakest predictor of attitudes toward vaccination).

At the time of the pandemic, another element is apparent that may play a significant role in reducing trust in science and its representatives—inconsistencies in the opinions they present. Since the emergence of COVID-19, discussion of the virus has been characterized by considerable disagreement among scientists and health experts observed on a number of issues, including what the origin of the coronavirus was, who is most vulnerable to infection and how effective certain treatments are [35,36]. However, according to the *experts-should-converge effect* [37], people expect experts to “speak with one voice” and divergence of opinions reduces their credibility. Shanteau [37] explains that since laypeople do not have enough knowledge to judge which of the presented positions is wrong, they may consider not trusting any of them as the most secure solution. Finally, the reason for the decline in trust in science may also be the suspicion that its representatives contributed to the outbreak of the epidemic. Despite the lack of grounds for such assumptions many people believe that the coronavirus is the result of laboratory manipulation or is created to gain profit from distributing new vaccines [38,39].

Trust in the Internet also appeared to be a strong predictor of the attitude toward vaccination in both measurements. However, in this case, the relationship was reversed. The more people trust the Internet as a source of health knowledge, the less positive their attitudes toward vaccination. There are two possible reasons for this. First, people who trust the credibility of information found on the Internet and search it for information about vaccinations may often end up with arguments spread by antivaccine activists which negatively influence their attitudes toward vaccination. It must be remembered that the Internet is blamed for the rise in vaccine skepticism. Due to widespread access to this powerful source of information, the arguments of anti-vaccine activists can be easily published and distributed, reaching many people and raising concerns regarding vaccination among some of them [22,40]. Moreover, the previous analyses of antivaccine content on the web showed that it offers a wide range of potentially attractive narratives that blend topics such as safety concerns, conspiracy theories and alternative health and medicine [41]. Second, it is also possible that the people who believe information obtained from physicians are different people than those who consider the internet to be a reliable source of health information. These people do not believe doctors and their recommendations, including regarding vaccinations, and they deliberately look for arguments supporting their doubts on the internet. What is interesting, our results showed that trust in the Internet was the only one of the factors included in the study that did not decrease in the second measurement carried out during the pandemic.

In our study, we investigated not only attitudes toward vaccination in general, but we also focused on the trust in different sources of health knowledge as possible predictors of willingness to be vaccinated against SARS-nCoV-2. However, in this case we found that the predictors we selected explained very little of the variation of the outcome variable. This result is very interesting because it shows that attitudes toward the new COVID vaccines may have different sources than attitudes toward vaccines that have long been known to the public. Our previous research (in preparation) shows that they are strongly associated with, among other things, the opinions people have about the pandemic (e.g., the belief that information about the threat is exaggerated in the media), the fear of being infected or the perceived likelihood of being infected.

### The Limitations of the Study

A limitation of our study may be the final number of subjects whose responses were analyzed. However, it is important to note that we used a longitudinal design and attempted to reach the same individuals almost three years later, which caused us to lose some respondents from the original sample. Additionally, with this design, we were able to see how attitudes had changed over such a long period of time (within subjects). Moreover, we were interested not so much in describing current attitudes toward vaccination in society (as there are many studies on this topic), but in examining how they are related to our predictors and how this relationship has changed over such a long period of time.

Another limitation of our research stems from the fact that when analyzing the predictors of people’s attitudes to vaccination (and changes in their opinion), we only took into account a few factors—belief in science and its achievements, belief in the correctness of experts’ opinions, and belief in the truthfulness of information that can be obtained from the Internet. While these variables were directly associated with the aims of our original 2018 study, there is no doubt that the list of factors which may serve as predictors of dynamics of human vaccination attitudes is much longer. First of all, in future research it is worth looking closely at the role of risk perception. While the research to date has considered the role of perceived risk in vaccination decision-making and vaccination attitudes [42,43], as far as we know, it has never been investigated how these opinions change over time as a function of perceived risk. Furthermore, we believe that both the perceived risk of getting sick and the risk of being vaccinated should be taken into account. On the basis of prospect theory [44] one may hypothesize that possible future losses (an undesirable post-vaccination reaction) may have more impact on attitudes towards vaccination than possible future gains (health). Another factor which should be taken into account is the social environment in which the person operates. If vaccination is the common norm in this community, refusing to vaccinate may run the risk of social exclusion. In a quite different social environment, however, a positive attitude towards vaccination may expose the person to the stigma of being a ‘weirdo’ [45]. Yet, another factor that may play an important role is related to individual differences. Two fundamental questions arise here. First, which personality traits are associated with attitudes to vaccination? Secondly, what traits are conducive to changing those attitudes? We intend to address these issues in our future research.

## Figures and Tables

**Table 1 vaccines-09-00933-t001:** Changes in trust towards different sources of health information (physicians, science, Internet, own knowledge) and change in attitudes towards vaccination—February 2018 vs. December 2020.

Variable	18 February	20 December	*F*	*p*	*Eta*
Trust in physicians	6.8	6.4	12.51	0.001	0.030
Trust in science	7.3	6.7	22.18	0.001	0.053
Trust in Internet	4.1	3.9	2.04	0.153	
Subjective knowledge	4.9	4.4	18.76	0.001	0.045
Attitudes towards vaccination	7.2	6.9	4.3	0.037	0.011

**Table 2 vaccines-09-00933-t002:** Regression analyses of attitudes towards vaccination as predicted by trust in different sources of health information (February 2018 and December 2020).

Variable	18 February			20 December		
	*B*	*SE*	*β*	*B*	*SE*	*β*
Trust in physicians	0.300	0.061	0.228 **	0.366	0.054	0.367 **
Trust in science	0.356	0.053	0.300 **	0.347	0.051	0.362 **
Trust in Internet	−0.267	0.048	−0.279 **	−0.255	0.050	−0.256 **
Subjective knowledge	−0.072	0.053	−0.066	−0.177	0.057	−0.160 *
*R* ^2^	0.34			0.49		
*F* for change in *R*^2^	51.71 **			99.78 **		

* *p* < 0.05, ** *p* < 0.001.

**Table 3 vaccines-09-00933-t003:** Changes in trust in different sources of health information as predictors of changes in attitude towards vaccination.

Variable	*B*	*SE*	*β*
Trust in physicians (change)	0.229	0.059	0.244 **
Trust in science (change)	0.266	0.058	0.288 **
Subjective knowledge (change)	−0.170	0.064	−0.160 *
*R* ^2^	0.17		
*F* for change in *R*^2^	28.49 **		

* *p* < 0.05, ** *p* < 0.001.

**Table 4 vaccines-09-00933-t004:** Trust in different sources of health information as predictors attitudes vaccination against SARS-nCoV-2 in December 2020.

Variable	*B*	*SE*	*β*
Trust in physicians	0.135	0.042	0.230 *
Trust in science	0.053	0.040	0.095
Trust in Internet	−0.107	0.038	−0.180 *
Subjective knowledge	−0.074	0.043	−0.113
*R* ^2^	0.12		
*F* for change in *R*^2^	15.19 **		

* *p* < 0.05, ** *p* < 0.001.

## Data Availability

The data and additional analyses are available at the OSF repository https://osf.io/eubzj/?view_only=0844820e79ce49fcb59e32c922e891a5, accessed on 20 June 2021.

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
