# Peer review of "The Credibility of Health Information Sources as Predictors of Attitudes toward Vaccination—The Results from a Longitudinal Study in Poland"

_vaccines, 2021, doi:10.3390/vaccines9080933_

Round 1

Reviewer 1 Report

The study is very current, but it follows an international and national scenario of large studies on the phenomenon, which are not mentioned. The analyzed panorama presents assertive data on the observed population, and references on the 'critical' aspects of online information are rarely used.

Then there are some questions, for example:

1) how were the interviewees reached to 'recompose' the sample?

2) was the first part of the study (2018) carried out in 'analogical' mode or always digital? Both studies were done with the web is not clear.

3) were there any reinforcing elements during vaccination?

4) references n. 40 cannot be used as it has not yet been published

Author Response

Dear Reviewer,

Following your message from July 22, 2021, we hereby submit a revised manuscript The credibility of health information sources as predictors of attitude toward vaccination – the results from longitudinal study in Poland. We thank the Reviewers for their helpful comments and have revised the text according to their requests. We trust that we have addressed all the criticisms. Detailed descriptions of the revisions and answers to the questions raised by Reviewer are listed below.

Answers to Reviewer 1

  1. The study is very current, but it follows an international and national scenario of large studies on the phenomenon, which are not mentioned. The analyzed panorama presents assertive data on the observed population, and references on the 'critical' aspects of online information are rarely used.

Ad.1      We have added a short paragraph and reference on critical aspects of online information (doubts about their reliability - which we think Reviewer had in mind). We addressed the studies on the trust to health care providers and the willingness to vaccinate against COVID19 in the end of the section 1.1. We would have preferred not to add a more extensive literature review and research findings, as this would have lengthened the introduction, which was already quite long and needed to be shortened (as was suggested by Reviewer 3).

  1. How were the interviewees reached to 'recompose' the sample?

Ad. 2 The both measurements of the study was conducted on the Ariadna panel, a nationwide research panel with over 300,000 Polish respondents that demographic profile corresponds to the profile of Polish internet users. A person who registers in the Panel receives his/her unique respondent identification number. This allow to return to the same individuals for longitudinal studies. We have added this information to the manuscript.

  1. Was the first part of the study (2018) carried out in 'analogical' mode or always digital? Both studies were done with the web is not clear.

Ad. 3 The both measures of the study (2018 and 2020) were online. Following Reviewer’s comment we clarify it in the manuscript (in Instruments section)

  1. Were there any reinforcing elements during vaccination?

Ad. 4. There were no any reinforcing elements during the vaccination

  1. References n. 40 cannot be used as it has not yet been published

Ad. 5 According to Reviewer’s comment we have removed the reference n.40

Reviewer 2 Report

This is a very interesting paper. The effect of knowledge and trust in vaccination is analyzed. It is necessary to consider the following points:

First, are there any independent variables other than trust and knowledge? It is necessary to consider independent variables such as perceived risk, benefit, and stigma.

Second, it is necessary to conduct regression analysis while controlling for demographic variables.

Third, it is necessary to describe the representativeness of the sample and the sampling error more specifically.

Author Response

Dear Reviewer,

Following the message from July 22, 2021, we hereby submit a revised manuscript The credibility of health information sources as predictors of attitude toward vaccination – the results from longitudinal study in Poland. We thank for your helpful comments and we trust that we have addressed all the criticisms. Detailed descriptions of the revisions and answers to the questions raised are listed below.

Answers to Reviewer 2

  1. Are there any independent variables other than trust and knowledge? It is necessary to consider independent variables such as perceived risk, benefit, and stigma.

Ad.1 In this research we focused only on the health information sources as independent variables. However, we agree with Reviewer that other variables (also perceived risk, benefit and stigma) should be considered (which we will certainly do when planning future studies). We appreciate this suggestion. We have also added the part related to this suggestion to the section on limitations of our study.

  1. It is necessary to conduct regression analysis while controlling for demographic variables.

Ad2. Following Reviewer’s suggestion we conducted the regression analysis with gender (dummy - coded) and age (due to it being an ordinal scale, we could not add education to the regression). Adding the demographic variables did not change the pattern of the results for the variables that we focused on. Therefore,  for clarity of presentation we did not include the results with the demographic in the manuscript. However, we added the regression results with demographics as supplementary material (that can be seen in OSF repository: https://osf.io/eubzj/?view_only=0844820e79ce49fcb59e32c922e891a5), we added also the detailed results of the repeated Anova for specific demographic variables.  

  1. It is necessary to describe the representativeness of the sample and the sampling error more specifically.

Ad. 3. Following Reviewer’s comments we described more briefly the representativeness of the sample in the manuscript. We have also added a detailed demographic description of the initial sample and the sample on which we finally performed our analyses as a supplementary materials.  In addition, we also conducted regression analyses with the same variables on the initial sample. The pattern of results was the same as in the final sample. We included a table with this analysis in the supplementary material.

Reviewer 3 Report

Thank you for the opportunity to revise this manuscript, which deals with a pivotal public health issue, i.e. the vaccination attitudes and the population (dis)trust and the vaccine hesitancy.

Unfortunately, even though the topic is paramount, the manuscript presents serious flaws and is not suitable for publication. In particular, the main problem is related to the evident selection bias of the sample analyzed, very limited if compared to the original sample observed in February 2018. Consequently, I believe the results are not significant and could not support the conclusions of the manuscript.

Furthermore, the quality of the presentation is very poor: the introduction section is very long and it often reports personal consideration of the authors, as well as the results section. The starting lines of the results section report the statistical methods and software used and must be moved in the right section.

Overall, there are several limitations in the present form of the manuscript related to the quality of the presentation and to the reliability of the results presented. For these reasons, I suggest a rejection.

Author Response

Dear Reviewer,

Following the message from July 22, 2021, we hereby submit a revised manuscript The credibility of health information sources as predictors of attitude toward vaccination – the results from longitudinal study in Poland. We thank for your helpful comments and we trust that we have addressed all the criticisms. Detailed descriptions of the revisions and answers to the questions raised are listed below.

Answers to Reviewer 3

  1. In particular, the main problem is related to the evident selection bias of the sample analyzed, very limited if compared to the original sample observed in February 2018. Consequently, I believe the results are not significant and could not support the conclusions of the manuscript.

Ad. 1.  We agree with Reviewer that the December 2020 sample is limited compared to the February 2018 sample. However, our goal was not to conduct a tracking study (in which the samples in each measurement of the survey are selected separately and their size and representativeness can be controlled), but a longitudinal study in which we examine if and how attitudes of the same respondents have changed over time. We wanted to find out how long-lasting are beliefs about different sources of information and, above all, how long-lasting are attitudes towards vaccination and whether such an unexpected situation as the COVID19 pandemic could change them. The two measurements were separated by a long time - almost three years, which greatly affected the sample reduction in the second measurement (in another longitudinal study we did, the gap was a few months and the attrition of subjects was small). We could, of course, have done the study in the first measurement with a larger sample, which would have simultaneously increased the likelihood of reaching more people in the second measurement, but at the time we did not anticipate that we would want to repeat it after such a long time (we did it because of the pandemic outbreak).

  1. Furthermore, the quality of the presentation is very poor: the introduction section is very long and it often reports personal consideration of the authors, as well as the results section.

Ad. 2 We agree with Reviewer that the introduction is quite long, we have tried to shorten it so as not to lose the clarity of the argument. Moreover, we have filled in the missing references, the absence of which could indeed suggest our personal considerations. The objection concerning personal considerations also referred to the results section, but it seems to us that we presented only the result of the analyses and it is not clear which exact fragments Reviewer had in mind.

  1. The starting lines of the results section report the statistical methods and software used and must be moved in the right section.

Ad. 3. Following Reviewer’s suggestion we have moved the information about the software and statistical methods to the Methods section.

Round 2

Reviewer 1 Report

In present form is soundness

Reviewer 3 Report

Thank you for the opportunity to review the revised version of the manuscript. Even though the authors have followed the suggestions made by the Reviewers, the selection bias of the sample does not allow to infer the results to the population.